# Suitability of Raycell MK2 Blood X-ray Irradiator for the Use in the Sterile Insect Technique: Dose Response in Fruit Flies, Tsetse Flies and Mosquitoes

**DOI:** 10.3390/insects14010092

**Published:** 2023-01-15

**Authors:** Hanano Yamada, Bénéwendé Aristide Kaboré, Nanwintoum Séverin Bimbilé Somda, Nonhlanhla L. Ntoyi, Chantel Janet de Beer, Jérémy Bouyer, Carlos Caceres, Robert L. Mach, Yeudiel Gómez-Simuta

**Affiliations:** 1Joint FAO/IAEA Centre of Nuclear Techniques in Food and Agriculture, International Atomic Energy Agency, Vienna International Centre, 1400 Vienna, Austria; 2Institute of Chemical, Environmental and Bioscience Engineering, Vienna University of Technology, Gumpendorfer Straße 1a, 1060 Vienna, Austria; 3Insectarium de Bobo-Dioulasso-Campagne d’Eradication de la mouche Tsétsé et de la Trypanosomose, Bobo-Dioulasso BP 1087, Burkina Faso; 4Unité de Formation et de Recherche en Science et Technologie (UFR/ST), Université Norbert ZONGO (UNZ), Koudougou BP 376, Burkina Faso; 5Vector Reference Laboratory, National Health Laboratory Services, Centre for Emerging Zoonotic and Parasitic Diseases, National Institute for Communicable Diseases, Johannesburg 2192, South Africa; 6Programa Operativo Moscas, IICA-SENASICA Km 19.5, Carretera Tapachula-Ciudad Hidalgo, Metapa de Dominguez 30860, Chiapas, Mexico

**Keywords:** *Ceratitis capitata*, *Anastrepha ludens*, *Glossina palpalis gambiensis*, *Aedes aegypti*, *Anopheles arabiensis*, X-ray, gamma ray, sterility, SIT

## Abstract

**Simple Summary:**

The sterile insect technique (SIT) is an environment-friendly, species-specific pest control method by which target insects are mass-produced in a factory and are made infertile by irradiation—usually with gamma rays. However, gamma sources are becoming more difficult and expensive to purchase, and the regulations surrounding these types of irradiators are becoming stricter. Therefore, there is now increasing interest in alternatives, such as X-ray irradiators. Following a recent technical evaluation of a blood X-ray unit, the aim of this research was to assess the biological responses of a selection of major SIT target insect species to irradiation in the X-ray unit as compared to gamma ray irradiation. It was found that all the insects responded similarly to X-rays as to gamma rays and that the X-ray unit is suitable for small- to medium-sized SIT programs.

**Abstract:**

The sterile insect technique (SIT) is based on the inundatory field release of a target pest following their reproductive sterilization via exposure to radiation. Until recently, gamma irradiation from isotopic sources has been the most widely used in SIT programs. As isotopic sources are becoming increasingly expensive, especially for small programs, and regulations surrounding their procurement and shipment increasingly strict, irradiation capacity is one of the limiting factors in smaller or newly developing SIT projects. For this reason, the possibility of using X-ray irradiators has been evaluated in the recent decade. The availability of “off-the-shelf” blood X-ray irradiators that meet the technical requirements for insect irradiation can provide irradiation capacity for those SIT projects in which the acquisition of gamma ray irradiators is not feasible. Following the recent technical characterization of a Raycell MK2 X-ray blood irradiator, it was found in this study, that MK2 instruments were suitable for the sterilization of fruit flies, tsetse flies and mosquitoes, inducing comparable, even slightly higher, sterility levels compared to those achieved by gamma ray irradiation. This, together with its estimated processing efficiency, shows that MK2 irradiators are suitable for small- to mid-sized SIT programs.

## 1. Introduction

Irradiation-induced sterilization of insects is an integral part of the sterile insect technique (SIT) [1] in which target pest species are produced in mass-rearing facilities, and males are made infertile before releasing them into a field site. Successful mating between the sterile males and wild females lead to a progressive decline in the target pest population density over successive generations and, thus, reduces crop loss and preserves animal, as well as human, health [1,2,3]. The Insect Pest Control Laboratory (IPCL) of the Joint FAO/IAEA Center of Nuclear Techniques in Food and Agriculture in Seibersdorf, Austria, houses several species and strains of fruit flies, tsetse flies and mosquitoes and has been driving research for the development of a SIT package against these insect pests of crops, livestock and human health. The most notable research toward the development or optimization of a SIT package against these pests in the past ten years was reviewed in [4].

The process of reproductive sterilization is one of the mandatory components of the SIT, and exposure to gamma radiation from isotopic sources is, to date, the most efficient, reliable and widespread method to achieve sterility in insects [5], especially in large SIT programs. However, the regulatory challenges and costs of procuring and transporting radioactive sources are very high and still rising, making the acquisition of 60Co-based irradiators a limiting factor for many SIT facilities, especially in the earlier phases of the projects [6]. The feasibility of using X-ray irradiators has been evaluated in the most recent decade, and it has been found that X-ray irradiation, in general, induces sterility in insects similarly to gamma ray irradiation in tested insects [7,8,9,10,11,12,13]. Furthermore, the availability of “off-the-shelf” blood X-ray irradiators with competitive purchasing costs that meet the technical requirements for insect irradiation can now provide irradiation capacity for those SIT projects in which the acquisition of gamma ray irradiators is not feasible. A recent technical characterization of a Raycell MK2 X-ray blood irradiator (Best Theratronics Ltd., Kanata, ON, Canada) showed that this irradiator provided a dose uniformity ratio of under 1.2, an average dose rate of 7.7 Gy/min, and 2 L of irradiation capacity [6], thereby meeting the FAO/IAEA-recommended minimum criteria for insect irradiation with X-rays [14]. Following this initial assessment, the current study aims to complete an evaluation of an MK2 instrument by providing biological dosimetry data in the form of dose response curves for select insect pest species. The Plant Pests, Livestock Pests and Human Disease Vector groups of the IPCL [4] perform three separate experiments to assess the suitability of MK2 irradiators for the sterilization of fruit flies, tsetse flies and mosquitoes, respectively. The experimental set-up reflects the insect species, strains and life stages, sample preparation and irradiation processes and doses used, as expected to be used in SIT programs. 

## 2. Materials and Methods

### 2.1. Irradiation Set-Up

All samples of insects were irradiated at a standard reference point, which was the center of a 2 L irradiation canister provided with the MK2 instrument used, as well as according to the dose rates and dose distribution map determined and described in Gómez-Simuta et al. [6]. Fruit fly and tsetse fly pupae were irradiated in instant rice for improved absorbed dose homogeneity in the sample and sample canister, as instant rice presents a similar density as insect pupae and, thus, serves as appropriate dummy material. Dose rates were, thus, measured in rice, and dose times were calculated accordingly. Mosquito pupae density is closer to water, whereas that of adults is closer to air. Dose time was, therefore, calculated according to dose rates measured in water and air, respectively. Where insects were additionally irradiated with a gamma ray irradiator for direct comparison, either a Foss Model 812 gamma irradiator (Foss Therapy Services Inc., North Hollywood, CA, USA) or a Gammacell 220 irradiator (Nordion Ltd., Kanata, ON, Canada) was used. These had dose rates of 56 Gy/min and 74 Gy/min, respectively.

### 2.2. Dosimetry

To ensure the accuracy of the irradiation dose given in each radiation event, Gafchromic HD-V2 or MD-V3 dosimetry films (International Specialty Products, Wayne, NJ, USA) were packed in small (2 × 2 cm) paper envelopes, which were placed near each insect sample. Gafchromic HD-V2 dosimetric films were previously indicated to be appropriate for the dose response of X-ray and gamma ray irradiation [6]. A DoseReader 4 instrument (Radiation General Ltd., Budapest, Hungary) appropriate for Gafchromic™ film [15] was used to read the films 24 h after irradiation. The standard operating procedure for Gafchromic™ film dosimetry [15] was followed to determine the absorption dose for each radiation event. The calibration used had a global uncertainty of 4.29%.

### 2.3. Dose Response of Ceratitis capitata and Anastrepha ludens Pupae under Hypoxic Conditions

#### 2.3.1. Strains and Rearing

The *Ceratitis capitata* VIENNA 8 genetic sexing strain (GSS) was developed at the IPCL [16]. This GSS was characterized by a pupal color mutation in which a wildtype copy of the markers was attached to the Y chromosome so that males expressed the wild phenotype (brown pupae), and females expressed the mutant phenotype (white pupae sensitive to temperature). Female embryos could be eliminated at the embryo stage by exposing the eggs to high temperature (34 °C) for 24 h [16]. The *Anastrepha ludens* GSS was developed in Mexico [17] and transferred to the IPCL in 2017. The males expressed a brown pupae phenotype, while the black pupae phenotype was expressed in females; then, the sexes could be separated at the pupal stage by using a pupal color-sorting machine.

The laboratory rearing conditions of the flies were 24 ± 1 °C, 60 ± 5% relative humidity (RH) and a photoperiod of 14 h light: 10 h dark. The adult flies were fed with a standard adult diet [18,19], which consisted of sugar and hydrolyzed yeast in a ratio of 3:1 and water ad libitum. The flies in this study were kept in 30 × 30 × 45 cm (length, width and height, respectively) cages that were covered with muslin cloth and had openings for experimental handling. The larvae of the flies were maintained on a carrot-powder-based diet. The pupae were separated according to the color of the pupa, as described above.

#### 2.3.2. Irradiation Procedure and Assessment of Sterility

*Ceratitis capitata:* Batches of brown pupae (males) two days before emergence were placed in plastic bags one hour before irradiation to achieve hypoxia, as described by Schwarz et al. [20] and the FAO/IAEA/USDA product quality control manual for fruit flies [21]. Two samples were irradiated separately in a Foss Model 812 Gamma Irradiator and in a Raycell MK2 irradiator at doses of 80, 90, 100, 125 and 145 Gy. Samples of pupae undergoing the same handling without irradiation were kept as controls.

After irradiation, pupae were kept in Plexiglas cages for fly emergence. Twenty-four hours after emergence, for each replicate, 20 sterile males and 20 fertile females were placed into a 30 × 30 × 30 plexiglass cage for sexual maturation, mating and oviposition. During the peak of the oviposition period, a sample of eggs was counted, placed on a piece of cloth, transferred to a larval diet and placed in an incubator at 28 °C for egg hatching. After 48 h, the number of unhatched eggs was recorded, and the number of pupae collected at the end of the experiment was registered. Five repetitions were performed for each dose and both the gamma and X-ray treatments.

*Anastrepha ludens*: Samples of *A. ludens* were collected, prepared and irradiated as described above for *C. capitata*. Only the dose of 80 Gy that is used in operational programs for the induction of reproductive sterility in this species was assessed for both the gamma ray (Foss Model 812) and X-ray (MK2) treatments. After irradiation, the pupae were returned to the insectary, and the dose response was assessed as described above for *C. capitata*. Nine repetitions were performed for each irradiator.

### 2.4. Dose Response of Glossina palpalis gambiensis Pupae

#### 2.4.1. Strain and Rearing

The *Glossina palpalis gambiensis* colony used in this assessment was established at the IPCL in 2009 from pupae derived from the Centre International de Recherche-Developpement sur l’Elevage en zone Subhumide (CIRDES) colony in Burkina Faso. Initially, the strain was colonized at Maisons-Alfort (France) in 1972 using pupae collected in Guinguette (Burkina Faso) and transferred to CIRDES in 1975 [22]. The last wild material introduced into the colony was collected at Mare aux Hippopotames in 1981. The colony, as well as the pupae and adults used in the assessment, were maintained at a constant temperature and relative humidity (RH) of 24 ± 0.5 °C and 75–80%, respectively, and under subdued and indirect illumination with a 12 h light: 12 h dark photoperiod [23,24]. The colony and experimental flies were fed three times per week on defibrinated bovine blood using an artificial membrane feeding system.

Pupae that were produced in the colony were collected daily and sex-sorted with a newly developed Infrared Pupae Sex Sorter (NIRPSS) at 23–24 days following larviposition. The NIRPSS was preconditioned with the following melanization parameters: T1 of 252, T2 of 0.10 and T3 of 10. The male pupae were selected from the same cohort of pupae classed as unmelanized when the unmelanized ratio (unmelanized pupae/total pupae sorted) was below 38%.

#### 2.4.2. Irradiation Procedure and Assessment of Sterility

Depending on the replication, fifty to seventy-five male *G. p. gambiensis* pupae were placed in a 60 mm × 13 mm petri dish without filling it, and this was placed in the middle of a cylindrical sample canister (2.0 l, 167 mm (Ø), 97 mm(H)) accompanying the Raycell MK2 instrument used. The remaining volume of the sample canister was filled with rice. The pupae were then exposed to radiation doses of 70, 90, 110 and 130 Gy. The control group was selected from pupae that were not irradiated. All irradiated and control pupae were handled and kept under similar conditions.

The irradiated and control pupae were incubated at 24 ± 0.5 °C and 75–80% RH until emergence. The teneral males were collected and kept in small cages (110 mm (Ø); 45 mm (H)) and fed as described above until sexual maturity. The females that emerged, due to a sorting error, were discarded. The seven- to eight-day-old irradiated and control males were mated in standard colony cages (Ø 20 cm) with three- to four-day-old virgin females at a 1:1 or slightly below male (*N* = 431): female (*N* = 592) ratio for four days, and their mortality was monitored daily. Males and females were then separated by chilling at 4 °C. The females were transferred to 20 cm diameter cages, and their daily production and mortality rates were recorded for 60 days. Six replications were performed for all doses.

### 2.5. Dose Response of Aedes aegypti and Anopheles arabiensis Pupae and Adults

#### 2.5.1. Strains and Rearing

The *Aedes aegypti* strain originated from field collections in Juazeiro (Bahia), Brazil, and was transferred to the ICPL from the insectary of Biofabrica Moscamed, Juazeiro, Brazil, in 2016. Both the *Aedes* strains were maintained following the “Guidelines for Routine Colony Maintenance of *Aedes* mosquitoes” [25]. The Dongola strain of *Anopheles arabiensis*, originating from Dongola, Northern State, Sudan, was donated by the Tropical Medical Research Institute, Khartoum, Sudan, in 2010 and was maintained at the IPCL following the anopheline mass-rearing guidelines [26].

#### 2.5.2. Irradiation Procedure and Assessment of Sterility

Eggs of *Aedes aegypti* from one cohort were collected and split in half to be hatched two days apart (one batch for collecting adults and one for collecting pupae for irradiation simultaneously). Males that emerged within an 8 h window were collected, counted into batches of 30, and placed in 15 × 15 × 15 cm Bugdorm^®^ cages (MegaView Science Co. Ltd., Taichung, Taiwan). The next day, the adult males were transferred to and irradiated in small 2 cL plastic cups closed with sponges. At the time of irradiation, the adults were 24–32 h old.

Pupae from the same cohort were collected in 4 h windows to ensure a uniform pupal age of 40–44 h, which is the most radioresistant age in this species. The pupae were sexed based on pupal size using a glass pupal sorter [27], and sex was verified under a stereomicroscope. All males were kept for irradiation, and females were transferred to individual tubes for emergence to ensure virginity for later mating. Male pupae were batched into groups of 30 in 2 cL plastic cups with excess water removed for irradiation.

The irradiation doses were selected according to the expected dose required to induce 50–100% sterility: 20, 55, 70 and 90 Gy. Both the pupae and adults in each technical repetition were irradiated simultaneously in a Raycell MK2 irradiator. Six repetitions were performed for all doses. Controls received the same handling but were not irradiated.

*Anopheles arabiensis* pupae were collected and sexed visually using a stereomicroscope. Females were placed in individual tubes for emergence to ensure virginity and were kept for later mating. Male pupae were counted into batches of 30 and were placed inside 2 cL plastic cups with excess water removed for irradiation. Male adults were knocked down in a cold room, counted into batches of 30 and placed into plastic tubes for irradiation. At the time of irradiation, male pupae were 24–28 h old, and adults were 24–30 h old. Both pupae and adults were irradiated in a Raycell MK2 irradiator with doses of 75, 90, 100, 110 and 120 Gy. Controls received the same handling but were not irradiated. Additional sample batches including controls were collected, sexed and prepared for irradiation with the same procedures but were irradiated in a GC220 gamma ray irradiator, with 55, 70, 95 and 110 Gy.

Following irradiation of both species, the males of each treatment group were placed in separate 15 × 15 × 15 cm Bugdorm^®^ cages. Thirty virgin females were added to each cage when the adults reached 2–3 days of age and were allowed to mate for 3 days before they were provided with 2 bloodmeals on consecutive days (days 6 and 7 after emergence). Oviposition cups were added to each cage on day 8 for mass egg collection (on days 9 and 10 after emergence) and were hatched following routine rearing protocols [25,26]. The total numbers of hatched and unhatched eggs were counted using a stereomicroscope. Any nonhatched eggs were either opened with a dissection needle, or if there were many, bleached to determine fertility status [28].

### 2.6. Statistical Analyses

The tsetse pupae emergence rate was analyzed using a generalized linear mixed model, where the dose was considered a fixed effect and the replicates as random effects. An emmeans comparison with the Tukey method was used to assess the differences between the irradiation dose treatments. The induced sterility of the tsetse flies was calculated by subtracting from 100% (pupae production in the control group) the treatment production relative to the control group, which was obtained by dividing the pupae produced in each irradiation dose treatment by the pupae produced in the control group.

Sterility in the fruit flies was calculated as the percentage egg hatch of the control group hatch rate. A Wilcoxon rank sum test with continuity correction was used to compare hatch rates of gamma- and X-ray-irradiated fruit flies. The residual fertility (RF) for mosquitoes was calculated as a proportion of the control fertility of each treatment group (RF = HRtx/HRc), where HRtx was the hatch rate of the treatment (tx) group, and HRc was the hatch rate of the control (c) group. Induced sterility (IS) was calculated by subtracting the RF from 1.

To analyze the dose response of pupae versus adults for *Ae. aegypti* and *An. Arabiensis*, a binomial GLMM fit by maximum likelihood (Laplace approximation) was used for egg hatch rates (considered as response variables), life stage (2 levels: pupae and adults) and irradiation log (dose) (4 levels: 20, 55, 70 and 90 Gy), and their interactions were considered fixed effects, with repetition as a random effect.

## 3. Results

### 3.1. Dosimetry

The dosimetry confirmed that all doses received lay within a 4.29% error range.

### 3.2. Sterilization Efficiency of Raycell MK2

All insect species used in this study responded to the X-ray irradiation in the MK2 instrument as expected, with induced sterility levels comparable to those achieved in alternative X-ray and gamma ray irradiators using the same doses. When comparing the dose responses in pupae of all five species, *Ae. aegypti* were the most radiosensitive, becoming nearly fully sterile (99.8% IS) at doses of 55 Gy and above. *A. ludens* were fully sterile at 80 Gy, whereas *An. arabiensis* and *C. capitata* showed similar dose response curves and needed at least 100 Gy to achieve above 99.9% IS. Finally, *G.p. gambiensis* needed a dose of 110 Gy or above to reach 99.6% IS (Figure 1).

#### 3.2.1. *Ceratitis capitata* and *Anastrepha ludens* Pupae

*Ceratitis capitata* pupae showed slightly higher levels of sterility (<2%) following irradiation with X-rays in the MK2 irradiator compared to irradiation with gamma rays (FOSS 812), (*p* = 0.036); a dose of 100 Gy resulted in 99.7% and 98.7% induced sterility (IS), respectively. Both the 125 Gy and 145 Gy doses gave full sterility, regardless of irradiator type. The dose responses following irradiation doses of 80, 90 and 100 Gy compared to the same doses of gamma irradiation are shown in Figure 2.

*Anastrepha ludens* irradiated as pupae in hypoxic conditions with 80 Gy with both X-rays (MK2) and gamma rays (FOSS 812) resulted in 100% sterility.

#### 3.2.2. *Glossina palpalis gambiensis* Pupae

After exposure to radiation, the pupae were incubated until emergence. A decrease in emergence rate from 89.7% to 83.8% was observed as the dose increased. A significant difference in the emergence rate was observed (X2 = 14.332, df = 4, *p* = 0.006) and was higher in pupae irradiated with 90 Gy compared to those irradiated with 110 and 130 Gy. The total number of eggs aborted was higher in females mated with irradiated males than those mated with fertile males at all doses (*p* < 0.001), and this was inversely correlated to the pupae production. The number of eggs aborted by females mated with males irradiated at 90 Gy and 110 Gy was higher than the number in females mated with males irradiated with 70 Gy. The fecundity of females mated with irradiated males decreased from 0.012 to < 0.001 as the irradiation doses increased from 70 to 130 Gy. In contrast, the mean induced sterility in females mated with irradiated males increased from 84.9 ± 6.7 to 95.7 ± 3.6, 99.6 ± 0.6 and 99.8 ± 0.5% as the irradiation doses increased from 70 to 90, 110 and 130 Gy, respectively, when using the MK2 instrument (Figure 3).

#### 3.2.3. *Aedes aegypti* and *Anopheles arabiensis* Pupae and Adults

As expected, the hatch rates of both *Ae. aegypti* and *An. arabiensis* reduced significantly with increasing dose (df = 4, *p* < 2.2 × 10^−16^). When irradiated in the MK2 instrument, *Ae. aegypti* male pupae and adults presented with sterility levels exceeding those observed following irradiation with gamma rays (Figure 4). In these species, no difference in radiosensitivity between the two developmental stages could be observed following exposures in both irradiators, as a dose of 55 Gy or above led to very high sterility of over 99.9% (Figure 4), (df = 1; *p* = 0.172).

Both male adults and pupae of *An. arabiensis* responded to X-ray irradiation in the MK2 irradiator similarly to gamma ray irradiation. Induced sterility was, however, slightly higher following X-ray irradiation (Figure 5). The adult stage in this species was also more radiosensitive than the pupal stage (df = 1; *p* = < 2.2 × 10^−16^), which corroborates data for the same strain irradiated with gamma rays (GC220) (Figure 5).

## 4. Discussion

The sterility data obtained from the five insect species tested in this report confirmed the relative biological effectiveness of MK2 irradiators compared to other X-ray and gamma ray irradiators. In the two tested fruit fly species, *C. capitata* was more sensitive following irradiation in the MK2 instrument than in the gamma ray irradiator, and *A. ludens* was fully sterile at the tested dose following irradiation in both irradiators. In a relevant study by Mastrangelo et al. [9], a (at the time) new generation of X-ray irradiator (RadSource2400) was evaluated in which the dose responses of *C. capitata* and *Anastrepha fraterculus* were assessed. It was also found that the exposure of the two fruit fly species to X-rays resulted in higher levels of sterility compared to gamma rays. In this case, 99% sterility in *C. capitata* was achieved with mean doses of 91.2 Gy with X-rays and 124.9 Gy with gamma rays, whereas 40–60 Gy was sufficient to sterilize *A. fraterculus* for both radiation treatments, which corroborates the results in this study. At present, most sterilization of insects is accomplished using gamma radiation, and considering that a dose of 80 Gy of gamma radiation was used in Anastrepha’s mass-rearing laboratories in some countries (such as Mexico and Guatemala) [5], our result of 100% sterility in *A. ludens* was achieved with 80 Gy of X-rays, which supports the suggestion that the use of X-rays could be an alternative technology for SIT with regard to biological dosimetry.

The MK2 irradiator was equally successful at sterilizing *G. p. gambiensis* pupae. Doses of 90 and 110 Gy were sufficient to induce 95.7 and 99.6% sterility in females that mated with exposed males. In other biological dosimetry tests for this species, gamma irradiation has been predominantly used for both adult and pupae sterilization, and for both life stages, a dose of 110 Gy only has induced sterility levels of 93.4% in adults [29] and 89.7% in pupae [30]. It has been reported in other studies that pupae are more sensitive to radiation than adults. When *Glossina brevipalpis* were treated as adults, a dose of 40 Gy induced 93% sterility in females, and the same dose when applied to pupae induced a sterility of 97% [31]. This variation in sensitivity between life stages was also seen for the subspecies of *Glossina palpalis palpalis* [32], for which a dose of 120 Gy was needed for adults and a dose of 60 Gy for pupae to induce sterility of 95%. The low induced sterility for pupae exposure to a dose of 110 Gy recorded by Ilboudo et al. [30] might be because of an error in the dose, as their dosimetry indicated an absorbed dose of 81 Gy. As an X-ray dose of 90 Gy was sufficient to induce sterility of 95.7%, further assessments to verify and compare the dose response of this species using two X-ray and one gamma ray irradiator are in progress.

In the two tested mosquito species, similarly, a lower X-ray dose was needed to reach the same level of sterility when compared to gamma irradiation, corroborating historical data where gamma ray irradiation has been applied. For *An. arabiensis* (Dongola strain), 110 Gy were required for >99% sterility in adults and 120 for pupae using the same Gammacell 220 machine with a cobalt-60 source and a dose rate of 16 Gy/min at the time of the experiment [33]. The GC220 irradiator was then introduced in 2010. Yamada et al. [34] observed higher residual fertility (14% and between 4 and 7%) in pupae of the same strain following irradiation in the same GC220 instrument with dose rates of 93 Gy/min (in earlier repetitions) and 84 Gy/min (in later repetitions), respectively [34]. In this study, using the same GC220 irradiator with a dose rate of 74 Gy/min at the time of the experiment, 110 Gy was sufficient to fully sterilize adults. However, pupae showed 4.6% residual fertility at the same dose. The sterility achieved in the MK2 irradiator in this particular strain of *An. arabiensis* showed fertility data lower than but closest to the data of Helinski et al. in 2006 [33]. This was likely because the dose rates of the MK2 are lower (average of 7.7 Gy/min) and closer to the GC220′s 16 Gy/min in 2006 than the other higher dose rates used in subsequent studies [35].

The dose response results for *Ae. aegypti* confirm former reports that *Aedes* spp. are generally more radiosensitive than Anophelines. In this particular experiment, male adults and pupae showed the same responses to the irradiation doses. However, a previous study showed that adults could be slightly more radiosensitive than pupae, although usually not significantly so [36]. This was not evident with the doses used in the MK2 irradiator, as doses of 55 Gy and above resulted in nearly full sterility. Other biological dosimetry tests performed in this strain of *Ae. aegypti* using a different X-ray irradiator (RS2400) with a dose rate of 9.11 Gy/min gave a very similar response curve (Yamada, unpublished data), whereas irradiation with gamma rays at lower dose rates induced higher sterility levels [37], and inversely, those with higher dose rates resulted in lower sterility levels [34].

These results, combined with the comparison of X-ray and gamma ray irradiation of *C. capitata*, support the findings of Yamada et al. [35], where it was shown that dose-dependent dose rate effects altered the dose response in mosquitoes and, likely, in other insects. At high doses, the higher the dose rate, the higher the residual fertility (and the lower the induced sterility). These new findings support our hypothesis that the increased sterilization efficiency of the X-ray irradiators is due to a dose rate effect. However, it is important to also investigate the effects of energy independent of dose and dose rate on insect dose response.

Apart from the relative biological effectiveness of an irradiator, processing efficiency is important for the assessment of its suitability for operational SIT programs. The requirements for this depend on the size and production capacity of the program and, thus, can vary. The largest SIT programs currently are those controlling fruit flies; the highest-producing facility is the El Pino facility in Guatemala, which has a production capacity of 3.6 billion sterile males per week. For the sterilization of such quantities, high-dose-rate, high-capacity, self-shielded or panoramic irradiators are needed. Other programs that require smaller production numbers, such as those in Hawaii, Costa Rica, Australia and many pilot facilities, can be run adequately with smaller irradiators, such as self-shielded GC220 instruments or, alternatively, blood X-ray irradiators such as the MK2 machine. Using the full volume of the 2 L canister, a fruit fly SIT program can sterilize approximately 13.4 million *A. ludens* and 25 million *C. capitata* pupae per week with one 8 h shift per day, with the potential to increase the processing capacity to 26.9 and 50 million per week, respectively, with the implementation of a second shift per day. Of course, these numbers can be increased by procuring more than one X-ray unit.

The current protocol of pupae irradiation that is used for the SIT program against tsetse flies in Senegal indicates that the pupae are irradiated inside specialized boxes designed for pupae shipment [38]. The recommended density of pupae inside a box is 1500 [38,39,40], and four boxes can fit inside the 2 L canister (i.e., 6000 tsetse pupae). Thus, with a processing time (exposure time for 110 Gy plus sample loading) of 19 min, three loads per hour can be irradiated. Therefore, around 1.1 million tsetse pupae can be treated in a 5-day week with two 6 h shifts. For the successful SIT program on Unguja Island (Zanzibar) in an area of 1650 km^2^, the largest number of flies that were ever released in one week was 102,557 [41]. Thus, the processing capacity of the MK2 irradiator more than meets the requirements of similar-sized tsetse SIT programs. If the full capacity of the 2 L canister were to be used, it could hold around 48,000 tsetse pupae. The output could, thus, be increased to around 8.6 million tsetse sterile males each week. However, the handling and packing protocols would need adjusting so as to not damage the pupae during irradiation. Additionally, in SIT programs against tsetse flies (and mosquitoes), blood used for feeding the colonies requires sterilization with irradiation at 1 k Gy to minimize contamination with pathogens (https://www.iaea.org/sites/default/files/guidelines-for-blood-processing-procedures.pdf, accessed on 16 September 2022). This can also be accomplished in an MK2 irradiator in just over 2 h.

For mosquito SIT programs against *Aedes* spp., in theory, around 75 million adult mosquitoes can be treated in a 5-day week with two 6 h shifts, as the full volume of the 2 L canister can hold 250,000 Aedes adults, and a processing time (exposure time for ~60 Gy plus sample loading) of 12 min allows for five loads per hour. The processing capacity of *An. arabiensis* would be less, as double the dose and, thus, double the time (~20 min) is needed for the exposure itself, thus allowing for only three loads per hour. Additionally, this species is slightly larger, and fewer adults can be compacted into the same space [42]. These numbers are adequate for current pilot SIT trials, the largest of which releases around 10 million sterile *Ae. albopictus* males per week. However, it is anticipated that large-scale area-wide control programs may require much higher irradiation capacity, for which multiple self-contained X-ray irradiators would be needed if it is not possible to house an industrial, high-throughput irradiator. It is also important to note that adult mosquitoes need to be immobilized by, for example, chilling [35,42,43] to be compacted in a container and not sustain injuries when moving around while packed. The chilling needs to remain for the duration of the irradiation exposure if irradiation times exceed 4–5 min, by which time adults start to become active at room temperature.

The MK2 irradiator, in common with other X-ray sources, has several advantages over isotopic sources: lower capital cost, much lower transportation costs and much simpler regulation and access control. As the generation of X-rays relies on electrical power, the radiation can also be easily turned off by removing the power. Servicing is more straightforward, as there is no radiation to contend with, and replacement tubes can be supplied by regular carriers. The supply of replacement cobalt-60 sources is both expensive and problematical. There have been an increasing number of cases of denial or delay of shipments of radioactive material [44,45,46,47,48], and there are stringent regulations and rising costs. The lower energy of X-ray systems means that it is much easier to block radiation, and typical X-ray systems are self-shielded and do not require a special room to house them. Finally, the skills needed for the handling of high-level radioactive sources are scarce, whereas the skills for handling the high-voltage systems required for X-ray are available in most countries.

The downside is, if the electrical supply is not reliable, the system does not function. All X-ray systems require good cooling to prevent the tubes from overheating, which can be difficult in remote locations, and X-ray tubes are rather fragile and susceptible to damage during transport. X-ray dose rates are often much lower than those from isotopic irradiators, and the lower energy and, in some cases, beam configuration restrict the volume that can be irradiated. In addition, X-ray systems are more likely to suffer failures due to the fragility of the tubes and the complexity of the electronics, high-voltage systems and external cooling units.

X-ray systems, therefore, offer advantages to small SIT programs with their lower costs and simpler regulation but are not yet able to compete with isotopic irradiators for larger programs. Although some SIT programs have implemented e-beam technology for high-throughput irradiation (for instance, 600 million *C. capitata* are irradiated per week in Spain), currently available industrial e-beam systems are very expensive, and purchase is not feasible for most facilities. Small, compact electron beam systems and flat panel X-ray technology show promise for the future but are not yet ready for use.

## 5. Conclusions

Overall, MK2 irradiators were suitable for the effective and reliable sterilization of three target insect groups of SIT. The observed biological responses to the X-ray irradiation were comparable to gamma ray irradiation—in this case, irradiation in an MK2 machine resulted in higher sterility levels than those obtained in the two tested gamma irradiators. Together with its good DUR and processing efficiency, the unit met the requirements for small- to medium-scale SIT programs for fruit flies, tsetse flies and mosquitoes. Further research on the effects of dose rate and energy can further the understanding of the differences between X-ray and gamma ray irradiation.

## Figures and Tables

**Figure 1 insects-14-00092-f001:**
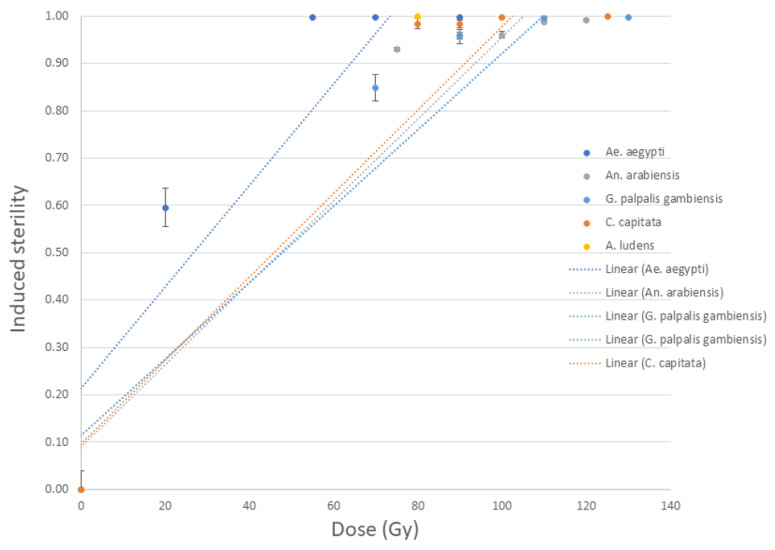
Induced sterility of pupae of five insect species in response to increasing irradiation doses in an MK2 irradiator. *A. ludens* was subjected to only one dose.

**Figure 2 insects-14-00092-f002:**
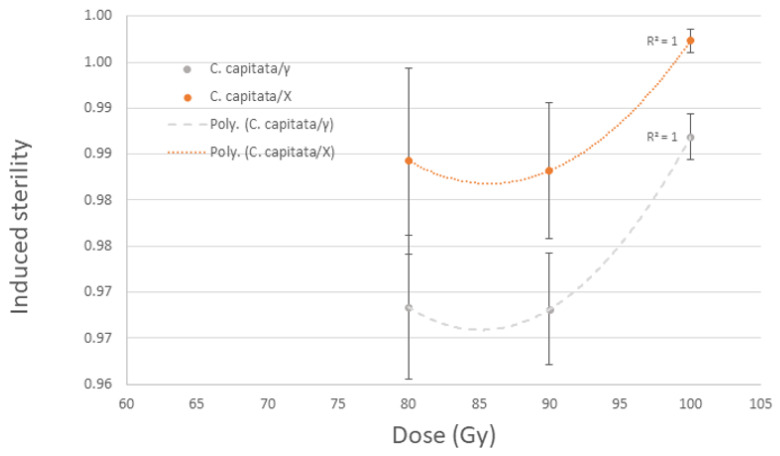
Dose responses of *C. capitata* following irradiation with 80, 90 and 100 Gy with X rays in an MK2 irradiator compared to γ-rays (Foss Model 812).

**Figure 3 insects-14-00092-f003:**
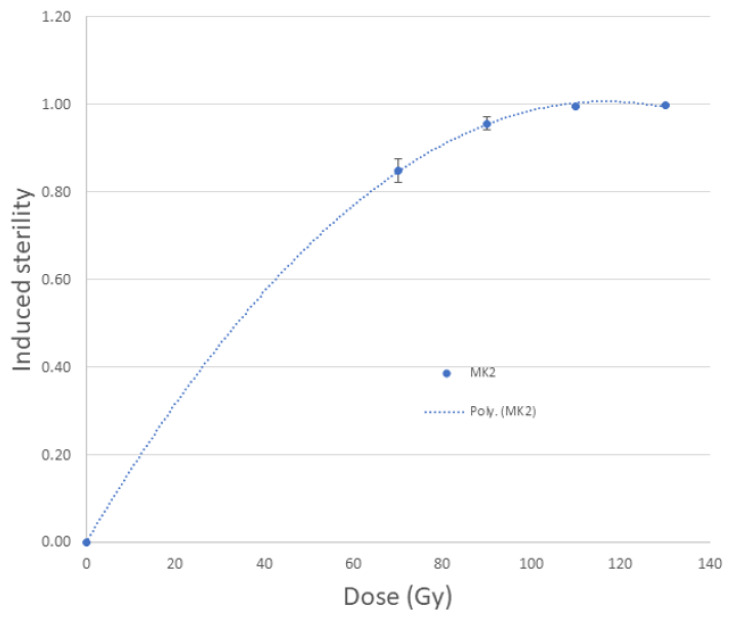
Dose response curve of *Glossina palpalis gambiensis* pupae irradiated with X-rays in an MK2 irradiator.

**Figure 4 insects-14-00092-f004:**
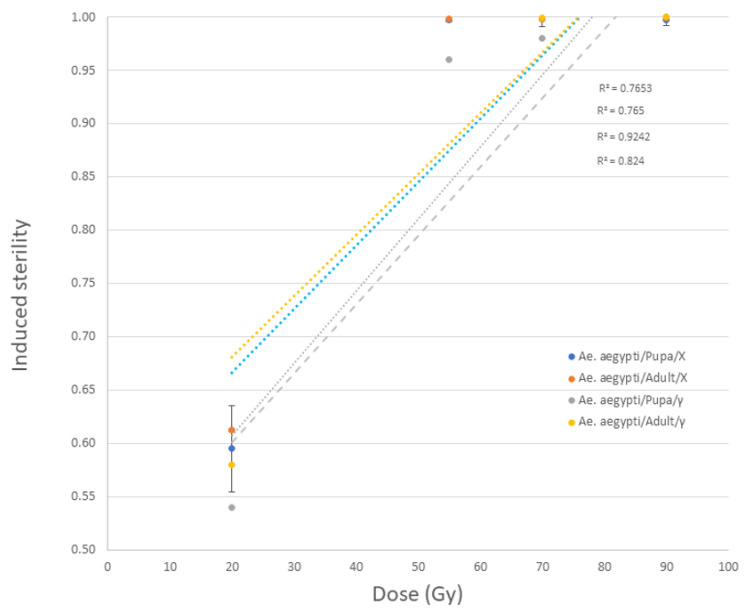
Dose response data of *Ae. aegypti* male pupae and adults irradiated with MK2 (X-ray) irradiator compared to the same strain irradiated in a GC220 instrument (gamma ray, (Yamada et al., 2022 [35])).

**Figure 5 insects-14-00092-f005:**
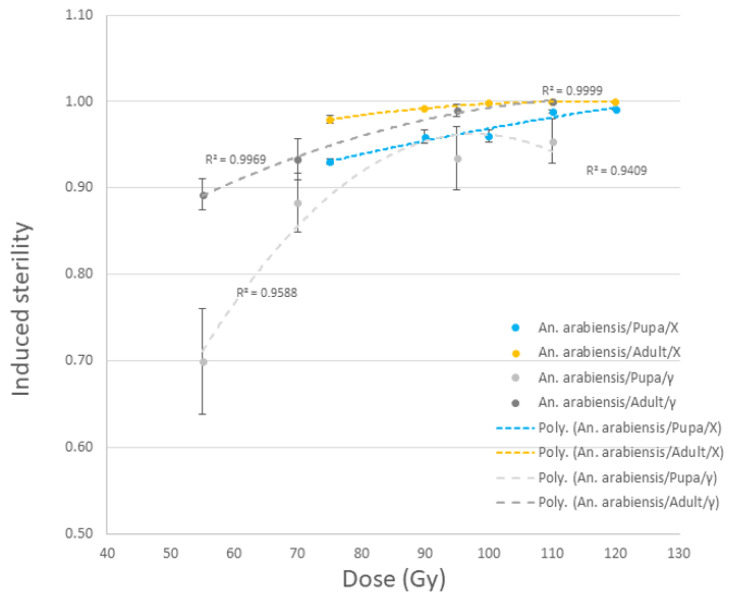
Dose response data of *An. arabiensis* male pupae and adults irradiated with MK2 (X-ray) irradiator compared to the same strain irradiated in GC220 irradiator (gamma ray).

## Data Availability

All original data can be supplied upon reasonable request.

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
