# Peer review of "Suitability of Raycell MK2 Blood X-ray Irradiator for the Use in the Sterile Insect Technique: Dose Response in Fruit Flies, Tsetse Flies and Mosquitoes"

_insects, 2023, doi:10.3390/insects14010092_

Round 1

Reviewer 1 Report

Title, lines 2-4, I suggest to change to a more concise title: "Suitability of the blood X-ray irradiator Raycell MK2 for the use in the sterile insect technique: dose response in fruit flies, tsetse flies and mosquitoes"

Line 80, Change "or" by "as"

line 229 delete the "e" from mass 

Line 236, use capital letter for Tukey

RESULTS

All the figures are of poor quality and need to be improved. The numbers and legends of the axes and notations need a larger font size. Delete the caption above the figure as it is inappropriate and repetitive with the caption below the figures.

lines 270-271, please joint both lines.

line 369, no italics in spp, and add a period at the end, and elsewhere ..,

line 376, to whom belongs the umpublished data?

line 398, change to A. ludens

REFERENCES

Lines 488-688. There are countless typos throughout the references. In scientific names, both words are capitalized, sometimes without italics; there are also typographical errors in the names of the journals, the names of the authors; some references are incomplete, etc., etc.  

Reviewer 2 Report

The article is well written and allows to clearly understand the topic and the novelty of the proposed approach. The experimental design is appropriated, data have been well analysed and results are interesting. However, in my opinion it would have been opportune to couple the data on the induced level of sterility to experiments regarding the male mating competitiveness at the different tested doses, as already done with other irradiation methods, otherwise it is difficult to exhaustively evaluate the method and its sustainability.

Anyway, I suggest just minor revisions to correct a few mistakes.

As an example, at line 262, arabiensis is written with a capital initial.

At line 324, "was more sterile" does non seem to me an accurate description. I suggest to replace it with something like "shows a higher level of induced sterility" or similar.

At line 339 (and elsewhere in the text), I suggest to not abbreviate the name of the species
